# Cerebellar Transcriptomic Analysis in a Chronic plus Binge Mouse Model of Alcohol Use Disorder Demonstrates Ethanol-Induced Neuroinflammation and Altered Glial Gene Expression

**DOI:** 10.3390/cells12050745

**Published:** 2023-02-25

**Authors:** Kalee N. Holloway, Marisa R. Pinson, James C. Douglas, Tonya M. Rafferty, Cynthia J. M. Kane, Rajesh C. Miranda, Paul D. Drew

**Affiliations:** 1Department of Neurobiology and Developmental Sciences, University of Arkansas for Medical Sciences, Little Rock, AR 72205, USA; knholloway@uams.edu (K.N.H.); jdouglas@uams.edu (J.C.D.); raffertytonyam@uams.edu (T.M.R.); ckane@uams.edu (C.J.M.K.); 2Department of Neuroscience and Experimental Therapeutics, Texas A&M University School of Medicine, Bryan, TX 77807, USA; marisa_pinson@tamu.edu (M.R.P.); rmiranda@tamu.edu (R.C.M.); 3Department of Neurology, University of Arkansas for Medical Sciences, Little Rock, AR 72205, USA

**Keywords:** AUD, astrocyte, microglia, oligodendrocyte, neuroinflammation, transcriptomics

## Abstract

Alcohol use disorder (AUD) is one of the most common preventable mental health disorders and can result in pathology within the CNS, including the cerebellum. Cerebellar alcohol exposure during adulthood has been associated with disruptions in proper cerebellar function. However, the mechanisms regulating ethanol-induced cerebellar neuropathology are not well understood. High-throughput next generation sequencing was performed to compare control versus ethanol-treated adult C57BL/6J mice in a chronic plus binge model of AUD. Mice were euthanized, cerebella were microdissected, and RNA was isolated and submitted for RNA-sequencing. Down-stream transcriptomic analyses revealed significant changes in gene expression and global biological pathways in control versus ethanol-treated mice that included pathogen-influenced signaling pathways and cellular immune response pathways. Microglial-associated genes showed a decrease in homeostasis-associated transcripts and an increase in transcripts associated with chronic neurodegenerative diseases, while astrocyte-associated genes showed an increase in transcripts associated with acute injury. Oligodendrocyte lineage cell genes showed a decrease in transcripts associated with both immature progenitors as well as myelinating oligodendrocytes. These data provide new insight into the mechanisms by which ethanol induces cerebellar neuropathology and alterations to the immune response in AUD.

## 1. Introduction

Excessive alcohol consumption in adolescents and adults has significant societal impacts, with an estimated economic cost of $249 billion in the U.S. alone [1]. Studies have shown that alcohol misuse can lead to low academic achievement, an increased risk of suicide, and a lifetime struggle with addiction [2,3,4]. Furthermore, alcohol use disorder (AUD) is one of the most prevalent mental health disorders, with 15.7 million Americans aged 12 and older diagnosed [5,6], and is associated with many physical and psychiatric comorbidities [7,8]. Despite the known consequences of excess alcohol consumption, 29.7% of men and 22.2% of women were diagnosed with an AUD in 2019 [9]. AUD is associated with pathology to organ systems including the central nervous system (CNS). Animal models of AUD have been developed which simulate the behavioral abnormalities and neuropathologies associated with human AUD, thus allowing researchers to investigate the biological mechanisms associated with AUD [10]. Within the CNS, the cerebellum is responsible for coordinating motor movements, cognitive processing, and sensory discrimination. In individuals with AUD, these cerebellar functions are often disrupted, which may persist following abstinence from alcohol [11,12]. Alcohol can induce an immune response in the CNS termed neuroinflammation, which may result in neurodegeneration [13] and an increased risk of developing an AUD [14]. In adult rodents, the extent of alcohol-induced neuroinflammation can depend on the experimental paradigm of ethanol exposure utilized [15,16,17,18,19].

In the current study, we evaluated the effects of ethanol on the transcriptomic profile of adult mouse cerebella, utilizing a chronic plus binge ethanol exposure paradigm adapted from an alcoholic liver disease model developed by the Gao laboratory, in which liver injury and systemic inflammation were reported [20,21]. Using a top-down approach, we analyzed the effects of ethanol on global gene expression in the cerebellum. Our studies indicated that ethanol altered the expression of immune-related transcripts and pathways in the adult cerebellum, and may alter the function and phenotype of CNS glial cells. Thus, the current studies aid in advancing our understanding of the neuroinflammatory transcriptomic changes induced in AUD, unraveling potential targets for therapeutic strategies.

## 2. Materials and Methods

### 2.1. Animals

All animal use protocols were reviewed and approved by the University of Arkansas for Medical Sciences (UAMS), Institutional Animal Care and Use Committee (IACUC). Adult C57BL/6J mice were purchased from The Jackson Laboratory (Bar Harbor, ME, USA; stock #000664) and were housed in the UAMS Division of Laboratory Animal Medicine, where a breeding colony was established to produce experimental animals. Adult male mice aged 10–14 weeks and weighing ≥20 g were housed individually and were randomly separated into 2 experimental groups, ethanol (E) or vehicle control (C), (*n* = 5 mice per group). Solid food was removed from cages, while water was provided ad libitum for the duration of the study. On study days 1–5, both experimental groups of mice were allowed to acclimate to the Bio-Serv Rodent Liquid Diet, control formulation (Flemington, NJ, USA; #F1259SP) provided freely in a fresh tube each day just before the start of the dark cycle. Following acclimation, the ethanol group underwent ethanol ramping, in which mice received successive increases of the Bio-Serv ethanol formulation (#F1258SP) with either 1% (day 6), 2% (day 7), or 3% ethanol (day 8) diluted using 95% *v*/*v* ethanol (Acros, a part of Thermo Fisher Scientific, Waltham, MA, USA; #AC615110010). On study day 9, chronic ethanol administration began, in which the ethanol-treated mice received 4% ethanol for 10 days, followed by 5% ethanol for 7 days. Pair-feeding for the control group began on study day 10 (the second day of 4% ethanol administration), in which the control group was fed an equivalent volume of control diet to match the mean ethanol group consumption volume from the previous day. On the morning of study day 26, immediately following the start of the light cycle, the ethanol group underwent an acute binge administration of 5 g/kg of 31.5% ethanol (*v*/*v*) diluted from 95% *v*/*v* ethanol delivered in water via gavage. The control group received 45% (*w*/*v*) Maltose Dextrin (10 DE Food Grade #3585) diluted in water and delivered via gavage. At this time, the liquid diet was removed from all cages and standard food pellets were provided. 24 h following the ethanol binge administration, mice were euthanized and transcardially perfused with 1X PBS containing 5 U/mL heparin. Brains were removed and cerebella were micro-dissected into two halves along the midline and snap frozen in liquid nitrogen. Blood ethanol concentrations (BECs) from a separate set of animals were determined to be 230 (±59.7) mg/dL following 4% administration, 311.7 (±49.8) following 5% administration, and 718 (±6.9) mg/dL following bolus administration, as reported previously when using this model [22]. BECs were not measured at the time of tissue collection, though we suspect BECs were at or near 0 based upon preliminary studies using this model.

### 2.2. Isolation of RNA, RNA-Seq Library Preparation, and Sequencing

One whole cerebellar hemisphere from each experimental animal was homogenized using a B2X24B Bullet Blender and 0.5 mm glass beads, as described by the manufacturer (Next Advance, Troy, NY, USA). RNA was isolated using the RNeasy Lipid Tissue Mini Kit with on-column Dnase digestion using the Rnase-free Dnase Set (Qiagen, Valencia, CA, USA, Cat #74804 and #79254), as described previously [23]. RNA quantity was assessed using the Qubit 3.0 fluorometer with the Qubit Broad-Range RNA Assay Kit (Thermo Fisher Scientific), and an Agilent Fragment Analyzer with the Standard Sensitivity RNA Gel Kit (Agilent Technologies, Santa Clara, CA, USA) was used to ensure RNA quality. RNA-seq libraries were prepared using an Illumina TruSeq mRNA Library Prep Kit with TruSeq Unique Dual Indexed adapters (Illumina, San Diego, CA, USA), and were quantified with Qubit 1X dsDNA High-Sensitivity NGS Gel Kit (Thermo Fisher Scientific). KAPA Library Quantification (Roche, Basel, Switzerland) was used for further library characterization, and an Agilent Fragment Analyzer with the High-sensitivity NGS Gel Kit (Agilent) was used for determining fragment size. Library molarities were calculated followed by dilution and denaturation according to manufacturer’s specification for clustering. The control and ethanol-exposed animals were clustered on a high-output NextSeq 500 flow cell and paired-end sequenced with 150-cycle SBS kit for 2X75 reads (Illumina).

### 2.3. Bioinformatic Analysis

To identify significant differences in mRNA gene expression and global biological pathways associated with alterations of cerebellar genes between the control and ethanol treatment groups, raw RNA-sequence data (NCBI GEO accession GSE222445) were analyzed. RNA-seq reads were quality-checked, trimmed, and aligned to the GRCm39 reference genome (accession: GCA_000001635.9) using the Nextflow RNAseq pipeline, nf-core/rnaseq (version 3.4), available at DOI 10.5281/zenodo.1400710. The resulting gene counts were transformed to Log_2_ counts per million (CPM) [24]. Lowly expressed genes were filtered out, and libraries were normalized by trimmed means of M-values [25]. The Limma R package was used to calculate differential expression among genes [26]. Log_2_ fold change values were calculated for ethanol compared to control, and genes with an adjusted (adj.) *p* ≤ 0.05 were considered statistically significant.

Heat map and principal component analysis (PCA) plots were created from the processed differential gene expression data using R statistical software. The R-based *EnhancedVolcano* package was used to make the volcano plots [27]. Pathway and network analysis were conducted using the QIAGEN Ingenuity Pathway Analysis (IPA) software (QIAGEN Inc., Valencia, CA, USA, https://digitalinsights.qiagen.com/IPA, accessed on 22 July 2022 ) using the “Core Expression Analysis”. IPA analysis parameters were set with the “species” parameter as “mouse”, and the “tissues and cell lines” parameter as “cerebellum”, with gene cut offs of an adj. *p* ≤ 0.05 and Log_2_ fold change ≥0.5 or ≤−0.5.

To obtain a better understanding of the specific cellular processes and cell types of the cerebellum that are most sensitive to ethanol exposure, we extracted cell type-specific gene lists from publicly available single-cell RNA-seq (scRNA-seq) resources, which have been used previously to deduce the cell composition of bulk RNA-seq tissue [28]. Using this approach, we identified a total of 822 microglia-associated genes from scRNA-seq resources [29,30,31,32,33] (Appendix A). We compared this list of microglia-associated genes to the list of genes significantly differentially regulated by ethanol (adj. *p* ≤ 0.05) in our dataset, which identified 151 microglia-associated genes whose expression was altered by ethanol (Table 1).

We were able to characterize 23 of the 151 genes as being either homeostatic or neurodegenerative (Table 2), as defined in previous studies [33,34,35,36,37] (Appendix A).

To further evaluate the effects of ethanol on homeostatic versus neurodegenerative microglial phenotypes, we computed mean z-scores to compare control versus ethanol for the transcripts associated with these phenotypes. Since the goal was to determine relative gene expression changes in our dataset, i.e., to determine whether the genes are up- or down-regulated due to ethanol, the average z-score was computed. We calculated the average z-score across individual genes in our extracted microglia homeostatic and neurodegenerative-associated gene lists, and then averaged these individual gene z-scores within each sample. The average z-score of each sample in the homeostatic and neurodegenerative group was then evaluated using a two-tailed Student’s t test, with *p* ≤ 0.05 being considered statistically significant. R statistical software was used to conduct the Student’s *t*-test as well as construct the average z-score graphs.

Similar to microglia, we utilized scRNA-seq data to compose a list of 309 astrocyte-associated genes (Appendix A) [37]. From this list we identified 56 astrocyte-associated genes that were differentially expressed in response to ethanol in our current study. We then characterized these transcripts as being associated with an astrocyte phenotype common to acute injury, chronic neurodegenerative diseases, or pan-injury (Table 3), the last of which includes genes associated with both acute injury and chronic neurodegenerative disease phenotypes [37].

To test for statistical significance, the average z-scores of each gene in our extracted acute, chronic, and pan-injury astrocyte-associated gene lists were generated, and these individual gene z-scores were then averaged within each sample in a manner consistent with the microglia described above. The Student’s *t*-test and average z-score graphs were constructed using R statistical software. Due to the small number of chronic neurodegenerative disease astrocyte-associated genes (*n* = 3), no z-score graph was generated for this group.

For oligodendrocyte lineage-associated genes, we extracted gene lists for oligodendrocyte precursor cells (OPCs) (381 genes), committed oligodendrocyte precursor cells (COPs) (55 genes), newly formed oligodendrocytes (NFOL) (9 genes), myelin-forming oligodendrocytes (MFOL) (347 genes), and mature oligodendrocytes (MOL) (7 genes) from publicly available scRNA-seq studies [29,30,39] (Appendix A), in a manner consistent with microglia and astrocytes described above, to determine which genes were significantly differentially regulated by ethanol. From these lists, we identified 71 differentially expressed genes associated with OPCs, 12 genes associated with COPs, 2 genes associated with NFOL, 2 genes associated with MOL, and 108 genes associated with MFOL within our significantly differentially regulated dataset (Table 4).

We performed statistical analyses in a manner similar to the microglia and astrocytes above. Briefly, the average z-scores of each gene in our OPC, COP, and MFOL-associated gene lists were generated, and the individual gene z-scores were then averaged between each sample. The Student’s *t*-test and average z-score graphs were constructed using R statistical software. Due to the small number of NFOL and MOL-associated genes differentially regulated by ethanol, z-score graphs were not generated for these groups.

## 3. Results

### 3.1. Alcohol-Induced Differential Gene Expression in the Cerebellum

A principal component analysis (PCA) was performed to provide an overview of the transcriptomic changes that occurred in response to ethanol. PCA analysis demonstrated that gene transcripts correlating and anticorrelating to the first and second principal components could differentiate control animals from those exposed to ethanol. (Figure 1A). Hierarchical clustering analysis of significant genes was conducted using Pearson’s correlation, while controlling for false discovery rate adj. *p* ≤ 0.05 (Figure 1B). RNA-seq analysis identified 732 genes that were significantly differentially regulated (adj. *p* ≤ 0.05 and log_2_FC 0.5). Of these 732 genes, 269 were upregulated genes (36.75%) and 463 were downregulated genes (63.25%), (Figure 1C).

### 3.2. Pathway Analysis of the Alcohol-Induced Differentially Regulated Genes

IPA analysis was performed to determine the specific pathways altered by ethanol in the cerebella of adult mice. The results of the top canonical pathways altered by ethanol exposure included those related to the generation of precursor metabolites and energy, pathogen-influenced signaling, cellular immune response, degradation/utilization/assimilation, cellular stress and injury, biosynthesis, disease-specific pathways, cardiovascular signaling, nuclear receptor signaling, and ingenuity toxicity list pathways (Figure 2A). A description of the pathway names, *p*-values, and molecules associated with each significantly altered pathway category is shown in Table 5. The top disease and biological function categories altered by ethanol exposure included nervous system development and function, tissue/cell morphology, cell-to-cell signaling and interaction, cell death and survival, cellular compromise, immune cell trafficking, and inflammatory response [−log(*p.val*) range = 5.5–2.1] (Figure 2B).

The diseases and biological function annotations that correlate to the diseases and biological functions categories, as shown in Figure 2B, are myelination (*p.val* = 2.88 × 10^−6^ ) or demyelination (*p.val* = 0.0053) of the cerebellum; quantity (*p.val* = 0.000125) or coupling (*p.val* = 0.000556) of oligodendrocytes; thickness of myelin sheath (*p.val* = 0.000556); quantity of cells (*p.val* = 0.00783); activation of microglia (*p.val* = 0.00783); permeability of blood–brain barrier (*p.val* = 0.0236); and astrocytosis of cerebella (*p.val* = 0.0467), (Table 6). These results suggest that in the cerebellum, ethanol alters biological functions that pertain to alterations in the formation of myelin, along with possible microglia and astrocyte phenotypic changes.

### 3.3. Alcohol Suppresses Microglia Homeostatic Genes while Increasing the Expression of Microglia Neurodegenerative-Associated Genes

Alcohol has been demonstrated to induce neuroinflammation in both humans and rodents which may include microglial activation, characterized by shortening and thickening of processes, along with the secretion of proinflammatory cytokines and chemokines that may contribute to neuropathology [19,40,41]. We performed hierarchical clustering analysis on homeostatic and neurodegenerative disease microglia-associated genes that were differentially expressed (adj. *p* ≤ 0.05) in response to ethanol (Figure 3A). A Student’s *t*-test comparing the average z-scores across all relevant genes indicated that ethanol caused an overall significant downregulation of microglia homeostatic genes (*p.val* = 3.191 × 10^−6^) (Figure 3B, Table 2) and an overall significant upregulation of microglia genes associated with neurodegenerative diseases (*p.val* = 7.786 × 10^−5^) (Figure 3C, Table 2). Collectively, these data suggest that ethanol may alter the microglial phenotype from a homeostatic and protective phenotype to a more activated phenotype observed in neurodegenerative diseases.

### 3.4. Astrocytes Undergo a Phenotypic Switch following Chronic plus Binge-like Alcohol Exposure

Astrocytes are one of the most abundant cell types in the CNS and play a critical role in regulating CNS functions in health and disease by maintaining homeostasis, providing energy to neurons, regulating synapse development and plasticity, modulating blood-brain-barrier integrity, and controlling neurological function and behavior [42,43,44,45,46]. Similarly to microglia, astrocytes play a role in CNS inflammation [47,48], and ethanol has been demonstrated to trigger an immune response in astrocytes [49,50]. In the current study, we performed hierarchical clustering analysis on acute injury, chronic neurodegenerative, and pan-injury astrocyte-associated genes that were differentially expressed (adj. *p* ≤ 0.05) in response to ethanol (Figure 4A). A Student’s *t*-test comparing the average z-scores across all relevant genes indicated that ethanol caused an overall significant increase in astrocyte genes related to acute injury (*p.val* = 7.085 × 10^−5^) (Figure 4B, Table 3) and an almost even number of up- and down-regulated genes (12 up vs. 13 down) pertaining to pan-injury (*p.val* = 0.6266) (Figure 4C, Table 3). Ethanol only altered the expression of three genes associated with the chronic neurodegenerative disease category (Table 3), thus the effect of ethanol on this small number of genes was not statistically evaluated. These data suggest that alcohol-induced transcriptomic changes in astrocytes are consistent with an acute injury phenotype.

### 3.5. Oligodendrocyte Lineage Cells Are Depleted upon Chronic plus Binge-like Alcohol Exposure

Ethanol has been demonstrated to alter myelination in adult humans and rodents [51,52]. We performed hierarchical clustering analysis on genes associated with distinct oligodendrocyte lineages (immature and myelinating) whose expression was altered by ethanol (Figure 5A,B). Evaluation of the effects of ethanol on immature oligodendrocyte lineages indicated that ethanol significantly decreased the expression of genes associated with COPs (*p.val* = 0.0006784) (Figure 5C, Table 4), and that ethanol skewed toward decreasing the expression of genes associated with OPCs (*p.val* = 0.1702) (Figure 5D, Table 4). For the myelinating oligodendrocyte lineage cells, ethanol significantly decreased the expression of genes associated with MFOLs (*p.val* = 2.905 × 10^−05^) (Figure 5E, Table 4). NFOL and MOL groups only contained two differentially expressed genes; therefore, statistical significance was not evaluated for these categories (Table 4). These results suggest that ethanol effects both immature and myelinating oligodendrocyte lineage cells, which could potentially lead to altered myelination.

## 4. Discussion

Pathway analysis indicated that ethanol had significant effects on immune processes in the cerebella of adult mice. In addition, these analyses suggested that ethanol may alter the phenotype and function of glial cells including microglia, astrocytes, and oligodendrocyte lineage cells. We and others have previously demonstrated that ethanol induces neuroinflammation in adult rodents. However, the amount of neuroinflammation varies depending on the ethanol administration paradigm. For example, acute 4-day ethanol exposure did not alter the expression of pro-inflammatory molecules, although microglial activation was observed [17,53]. Following 10-day ethanol exposure, increased expression of pro-inflammatory molecules was observed, although it was somewhat modest [16,18,19]. Chronic ethanol exposure over a period of 3–5 months resulted in more robust neuroinflammation [15,49,54,55]. Using a variation of the same model as the current study, in which gene expression in both male and female mice was evaluated in control, ethanol, and ethanol + pioglitazone experimental groups, we have previously demonstrated robust neuroinflammation following chronic plus binge exposure to ethanol in less than one month [22]. This model is similar to an alcoholic liver disease model used previously by the Gao laboratory, in which they showed systemic inflammation and liver injury [20,21]. At this point, it is unclear in our studies whether ethanol induces CNS inflammation directly or indirectly through ethanol induced inflammation outside of the CNS. In order to begin to understand the possible mechanisms by which ethanol induces neuroinflammation in this chronic plus binge model of AUD, we have treated a unique set of male mice for the purpose of RNAseq analysis in the current study. We acknowledge that the use of only male mice is a limitation of the current study. Furthermore, some of the pathways identified in the current study only contain 1 or 2 genes, and some genes are represented in multiple pathways. Thus, we have exercised caution to not overinterpret the results.

We evaluated the transcriptomic data to identify immune-regulated genes whose expression was most strongly induced by ethanol, which included *FOSB*, *CCL2*, *CCL7*, *C5AR1*, *SPP1*, *CD68*, *SOCS3*, *C3AR1*, and *KLF4.* The most highly upregulated gene is *FOSB*, which encodes a transcription factor that dimerizes with Jun protein to form AP-1 and plays a critical role in alcohol and drug addiction [56]. Alcohol increases the expression of *FOSB* in the mesocorticolimbic system, which is believed to contribute to alcohol use disorder [57,58]. Furthermore, ethanol was demonstrated to alter synaptic plasticity and epigenetic alterations in the *FOSB* promoter, resulting in increased *FOSB* expression in the medial prefrontal cortex in wild-type but not *TLR4* deficient mice. Since ethanol is believed to activate TLR4, resulting in downstream immune signaling [59], a role of ethanol-induced neuroinflammation is suggested in these processes. *FOSB* has also been demonstrated to contribute to excitotoxic microglial activation through regulation of complement C5a receptors in these cells [60]. Interestingly ethanol strongly increased the expression of complement *C5AR1* and *C3AR1* in our RNA-Seq studies. *C5AR1* expression is increased in the liver of patients with alcoholic hepatitis [61], and is believed to contribute to alcohol-induced inflammation and liver injury [62,63]. Additionally, ethanol induces the expression of complement receptors including *C3AR1* expression in microglia, resulting in altered phagocytosis [64]. We previously demonstrated that ethanol induces the expression of the chemokine *CCL2* or *MCP-1* following acute ethanol exposure in adult rodents [65], as well as in animal models of fetal alcohol spectrum disorders (FASD) [66]. It is interesting that in the current study, ethanol induced the expression of *CCL2* as well the related chemokine *CCL7* or *MCP-3* in this chronic plus binge model. It should also be noted that transcriptomic changes were only evaluated at one timepoint, 24 h after the final ethanol exposure. Future studies may wish to evaluate transcriptomic changes at different times following the final ethanol exposure. It is also noteworthy that the other immune-related molecules we identified previously in this model were not indicated in the current study; this may be due to less sensitivity and smaller “n”, both of which are limitations that come with RNAseq when compared to quantitative real-time PCR [22].

Microglia are capable of responding to signals, resulting in activation and an altered phenotype. Our IPA analysis indicated that ethanol treatment resulted in microgliosis or microglial activation in the cerebellum. Upon activation, microglia have traditionally been hypothesized to undergo classical activation, resulting in a M1 pro-inflammatory phenotype, or alternative activation, resulting in an M2 anti-inflammatory or protective phenotype [67,68]. However, more recently it has become clear that microglial phenotypes are complex, and cannot be defined or categorized effectively using this simple binary system [69]. One recent nomenclature to distinguish microglial phenotype focuses on homeostatic versus neurodegenerative disease phenotypes. Under homeostatic conditions, microglia have a homeostatic phenotype, described by playing a role in synaptic plasticity and synaptogenesis, trophic support, chemotaxis and immune cell recruitment, and neurogenesis [37]. During insult to the CNS, microglia commonly lose their homeostatic signature and assume a chronic inflammatory signature [70,71,72]. Evaluation of the phenotype of microglia in a variety of neurodegenerative diseases have resulted in the identification of a common neurodegenerative disease-associated microglia phenotype [34,37,71,73]. In the current study, ethanol induced a microglia phenotypic switch in the cerebellum. This phenotypic switch was similar to that observed in neurodegenerative diseases, with a downregulation of homeostatic signature genes and an upregulation of neurodegenerative signature genes.

Astrocytes, like microglia, are capable of functioning in the innate immune response in the CNS. Once astrocytes are activated, commonly referred to as astrogliosis/astrocytosis, they produce cytokines and chemokines, nitric oxide, and other reactive oxygen species as part of an inflammatory response [74], Our IPA analysis indicated that ethanol treatment resulted in “astrocytosis”. Astrocytes were classically defined to respond to various stimuli to become reactive A1 astrocytes (neurotoxic or reactive A2 astrocytes) which are protective and neurotrophic [75,76]. However, as with microglia, this binary system of classifying reactive astrocytes appears inadequate to fully define and distinguish astrocyte phenotypes. More recently, Serrano-Pozo and colleagues performed a meta-analysis of mouse transcriptomic studies which resulted in a nomenclature that classified reactive astrocytes as being consistent with acute injury, chronic neurodegeneration, or pan-injury reactive astrocytes which exhibited characteristics of both acute injury and chronic neurodegenerative phenotypes [38]. In the current study, we determined that ethanol induced changes consistent with an acute injury astrocyte phenotype. Interestingly, LPS was previously shown to trigger an acute injury astrocyte phenotype [38]. ethanol has also been shown to activate TLR4 receptors, suggesting that ethanol-mediated neuroinflammation could occur in response to recruitment of TLR4 during alcohol use/abuse [77,78,79]. Therefore, we speculate that in this model of AUD, in the cerebellum, ethanol induces an acute injury astrocytic phenotype through the activation of TLR4, subsequently inducing an immune response.

Oligodendrocytes are responsible for forming a myelin sheath around axons of neurons in the CNS, facilitating the efficient propagation of action potentials [80]. OPCs are produced during embryogenesis, and migrate to their functional location wherein they differentiate into mature myelinating oligodendrocytes. Most myelination occurs at later stages of CNS development but can occur throughout life [81]. Ethanol has profound effects on the developing CNS and is believed to significantly contribute to the pathology associated with FASD, at least in part by altering myelination [82]. Ethanol also alters myelination in adults with AUD [83,84]. Ethanol is highly toxic to oligodendrocyte lineage cells, with OPCs being particularly susceptible [85,86]. Alcohol exposure is known to disrupt OPC differentiation and survival by decreasing the expression of platelet-derived growth factor receptor α (PDGFRα), a molecule crucial for differentiation of OPCs into mature oligodendrocytes [87]. In the current study, we found that adult chronic plus binge-like alcohol exposure depletes the expression of genes associated with both immature oligodendrocyte precursor cells as well as myelinating oligodendrocytes. Future studies are needed to determine the mechanism by which ethanol effects oligodendrocyte lineage cells and myelination in AUD.

## 5. Conclusions

The current study demonstrates that ethanol alters the transcriptomic profile in the adult cerebellum in a chronic plus binge model of AUD. The pathways altered by ethanol included those involved in immune response. Ethanol caused a shift in the expression of microglial-associated genes, with a decrease in homeostatic and an increase in chronic neurodegenerative-associated transcripts. Ethanol also increased the expression of astrocyte-associated genes common to acute injury. Finally, ethanol decreased the expression of genes associated with immature oligodendrocyte progenitor cells, as well as myelinating oligodendrocytes. These results provide clues about the mechanisms by which ethanol induces neuroinflammation and altered glial function in AUD.

## Figures and Tables

**Figure 1 cells-12-00745-f001:**
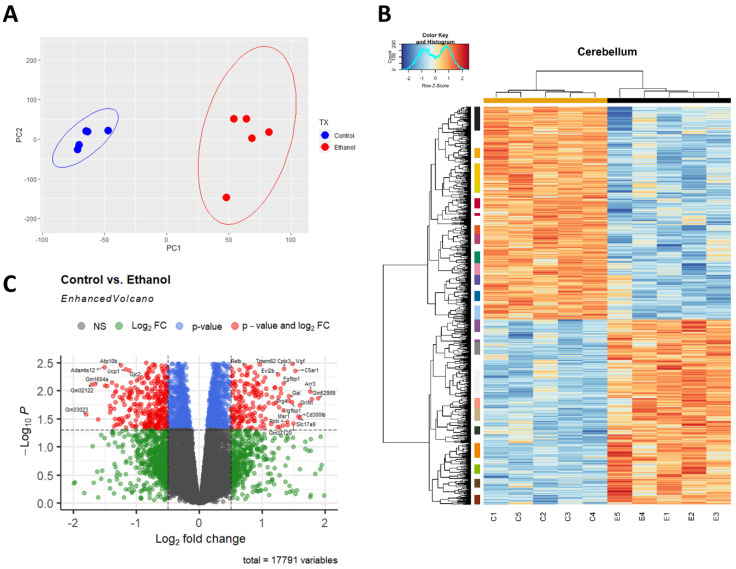
Ethanol-induced differential gene expression in the cerebellum. Principle component analysis (PCA) of genes contributing to variance between ethanol (E) and control (C) in the cerebellum were analyzed using R statistical software (**A**). A heatmap and hierarchical clustering dendrogram of relative gene expression across samples was constructed using R statistical software for significantly (adjusted *p* < 0.05) altered genes. Red indicates positive z-scores (upregulation) and blue indicates negative z-scores (downregulation) (**B**). The R *EnhancedVolcano* package was utilized to construct a volcano plot displaying fold change versus adjusted p-value of all detected genes in the cerebellum. 732 of 17,791 total identified transcripts displayed an adjusted *p* < 0.05 and Log_2_ fold change ≥0.5 or ≤−0.5, shown in red (**C**). *n* = 5 males per treatment group E or C.

**Figure 2 cells-12-00745-f002:**
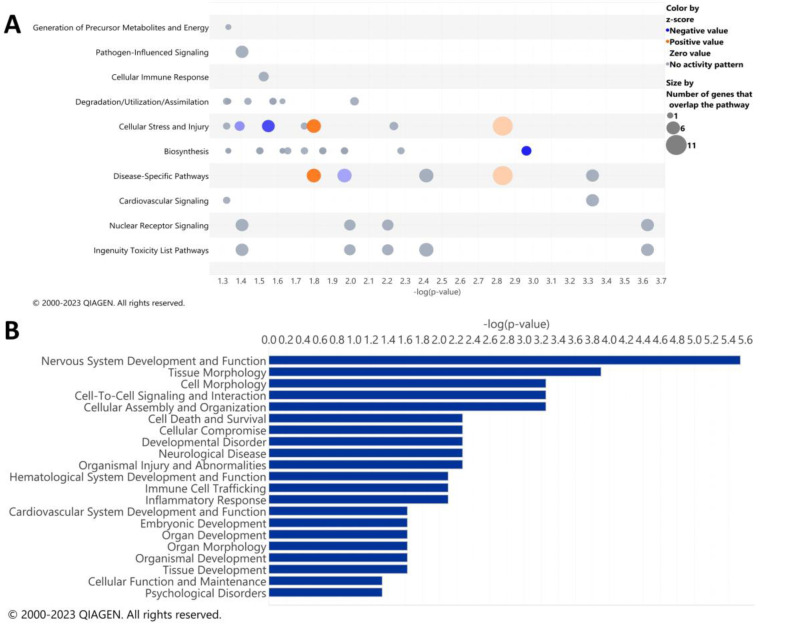
Top canonical pathways and top diseases and biological functions in the cerebellum altered by ethanol exposure. Qiagen Ingenuity Pathway Analysis (IPA) software was utilized to assess the top canonical pathways (**A**) and the top diseases and biological functions (**B**) altered by ethanol exposure using the “cerebellum” selected analysis settings. All analyses were restricted to genes with an adjusted *p* < 0.05 and Log_2_ fold change ≥ 0.5 or ≤−0.5. *n* = 5 males per treatment group E or C.

**Figure 3 cells-12-00745-f003:**
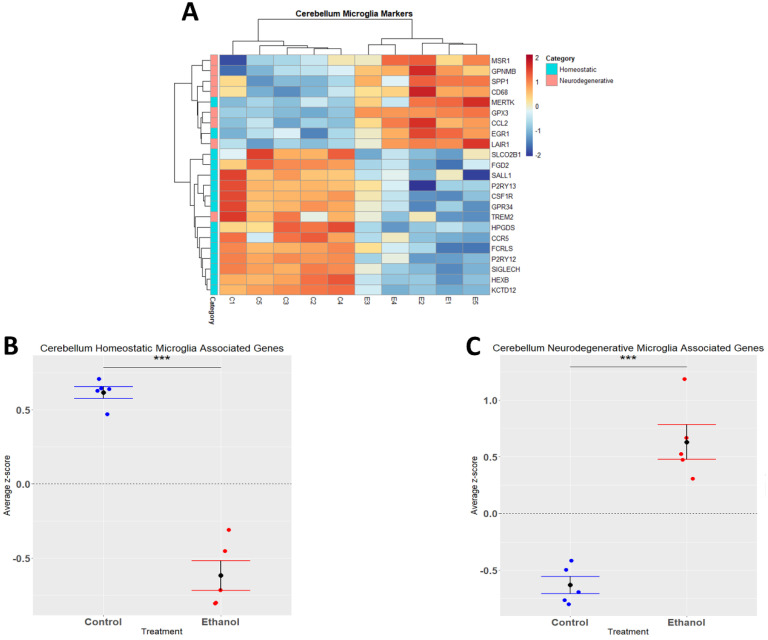
Microglia-associated genes altered by ethanol exposure in the cerebellum. R statistical software was utilized to construct a heatmap and hierarchical clustering dendrogram of relative gene expression across samples for significantly (adjusted *p* < 0.05) altered and categorized microglia-associated genes as detailed in Methods. Red indicates positive z-scores (upregulation) and blue indicates negative z-scores (downregulation) (**A**). Individual genes were z-scored across samples, followed by calculation of average z-score for each treatment group which was used for testing statistical significance in R with Student’s *t*-test. Quantification by average z-score of homeostatic microglia-associated genes (**B**) and neurodegenerative microglia-associated genes (**C**). *n* = 5 males per treatment group E or C; *** *p* < 0.001.

**Figure 4 cells-12-00745-f004:**
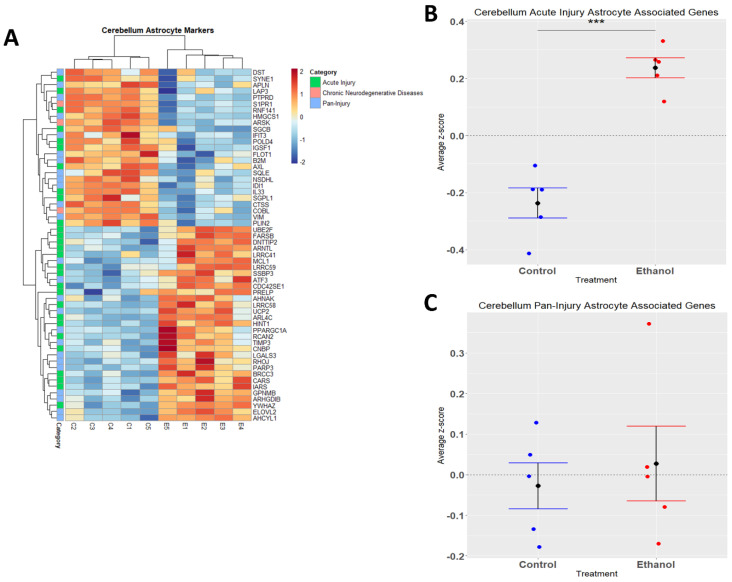
Astrocyte-associated genes altered by ethanol exposure in the cerebellum. R statistical software was utilized to construct a heatmap and hierarchical clustering dendrogram of relative gene expression across samples for significantly (adjusted *p* < 0.05) altered and categorized astrocyte-associated genes, as detailed in Methods. Red indicates positive z-scores (upregulation) and blue indicates negative z-scores (downregulation) (**A**). Individual genes were z-scored across samples, followed by calculation of the average z-score for each treatment group, which was used for testing statistical significance in R with Student’s *t*-test. Quantification by average z-score of acute injury astrocyte-associated genes (**B**) and pan-injury astrocyte-associated genes (**C**). Due to the small number of chronic neurodegenerative injury astrocyte-associated genes, no z-score graph was generated for this group; however, this group is further characterized in Table 3. *n* = 5 males per treatment group E or C; *** *p* < 0.001.

**Figure 5 cells-12-00745-f005:**
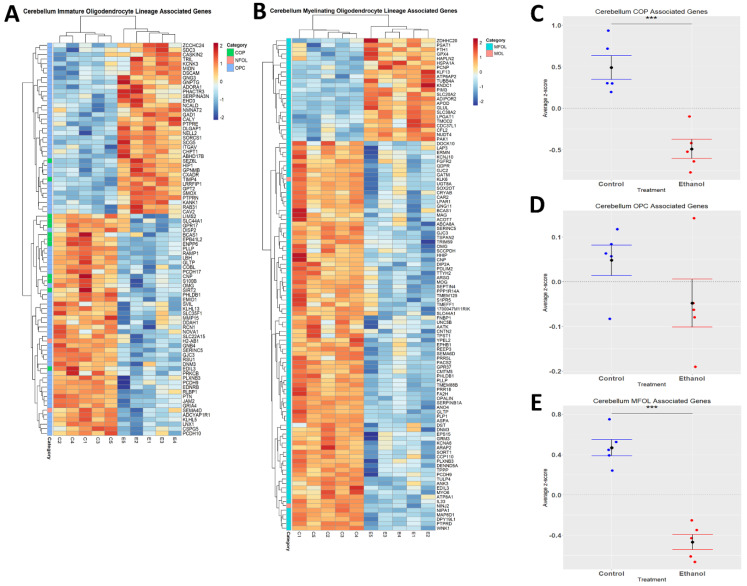
Alterations in oligodendrocyte lineage-associated genes by ethanol exposure in the cerebellum. R statistical software was utilized to construct a heatmap and hierarchical clustering dendrogram of relative gene expression across samples for significantly (adjusted *p* < 0.05) altered and categorized oligodendrocyte lineage-associated genes as detailed in Methods: immature oligodendrocyte lineage-associated genes (**A**) and myelinating oligodendrocyte lineage-associated genes (**B**) Red indicates positive z-scores (upregulation) and blue indicates negative z-scores (downregulation) (**A**,**B**). Individual genes were z-scored across samples, followed by calculation of average z-score for each treatment group, which was used for testing statistical significance in R with Student’s *t*-test. Quantification by average z-score of COP-associated genes (**C**), OPC-associated genes (**D**), and MFOL-associated genes in the cerebellum (**E**). Due to the small number of NFOL and MOL-associated genes, no z-score graph was generated for this group; however, this group is further characterized in Table 4. Abbreviations: OPC, oligodendrocyte precursor cell; COP, committed oligodendrocyte precursor; MFOL, myelin-forming oligodendrocyte; NFOL, newly formed oligodendrocyte; MOL, mature oligodendrocyte. *n* = 5 males per treatment group E or C; *** *p* < 0.001.

**Table 1 cells-12-00745-t001:** Uncategorized microglia-associated genes dysregulated by ethanol exposure in the cerebellum. Genes were identified by cross-referencing our significantly (adjusted *p* < 0.05) differentially regulated gene list with the 822 microglia-associated genes extracted from previous studies [identified in [29,30,31,32,33]] (Appendix A) using R statistical software, which identified 151 genes associated with microglia.

Symbol	LogFC	Adj. *p*	Symbol	LogFC	Adj. *p*	Symbol	LogFC	Adj. *p*	Symbol	LogFC	Adj. *p*	Symbol	LogFC	Adj. *p*
FOSB	2.81	0.0081	IFRD1	0.65	0.0001	SLC25A5	0.27	0.0010	CMTM6	−0.21	0.0438	PIK3CD	−0.50	0.0057
GPX3	2.68	1.77 × 10^−9^	ZFP36	0.62	0.0027	CCNL1	0.27	0.0035	MKNK1	−0.22	0.0422	CTSS	−0.51	0.0005
CCL2	2.44	0.0015	KLF4	0.60	0.0238	FTL1	0.26	0.0021	EDEM2	−0.23	0.0235	PLD4	−0.52	0.0208
CDKN1A	2.31	0.0007	ANXA3	0.58	0.0021	TMSB4X	0.26	0.0037	DOCK10	−0.23	0.0350	KCTD12	−0.53	1.79 × 10^−5^
FCNA	2.06	0.0028	ARHGDIB	0.54	0.0103	PTBP1	0.23	0.0289	RGS3	−0.23	0.0465	IFI203	−0.54	0.0313
MAFF	1.94	0.0002	IER3	0.50	0.0012	MYLIP	0.23	0.0321	TLN2	−0.24	0.0188	COL27A1	−0.54	0.0433
CCL7	1.81	0.0025	IER2	0.50	0.0318	BRD2	0.23	0.0038	SLC38A6	−0.24	0.0467	HPGDS	−0.60	0.0100
C5AR1	1.53	0.0044	PROS1	0.48	0.0116	KLF6	0.23	0.0368	PLXDC2	−0.24	0.0134	UNC93B1	−0.60	0.0014
GM3002	1.40	0.0405	ICAM1	0.46	0.0449	MCL1	0.21	0.0160	RGL2	−0.25	0.0089	TREM2	−0.62	0.0170
MSR1	1.34	0.0221	CFH	0.45	0.0092	PCF11	0.21	0.0071	PPCDC	−0.25	0.0401	ITGAM	−0.65	0.0010
EVI2B	1.25	0.0051	LAIR1	0.45	0.0055	CLTC	0.21	0.0070	SLC29A3	−0.25	0.0314	CCR5	−0.67	0.0274
LYVE1	1.22	0.0164	DUSP6	0.44	0.0070	CYFIP1	0.20	0.0136	ZFP90	−0.25	0.0257	SELPLG	−0.67	0.0003
UCP2	1.20	0.0088	REL	0.44	0.0343	ZCCHC2	0.20	0.0245	SLCO2B1	−0.28	0.0484	DSN1	−0.68	0.0116
CSRNP1	1.10	8.39 × 10^−6^	RGS2	0.43	0.0281	FMNL1	0.19	0.0425	CAMK1	−0.28	0.0040	IRF7	−0.70	0.0273
APOC1	1.05	0.0009	TSPO	0.42	0.0433	SERINC3	0.19	0.0467	GPR155	−0.28	0.0130	APOBEC1	−0.70	0.0296
SPP1	1.05	0.0315	ZFP36L2	0.41	0.0021	IL16	0.18	0.0149	TLR3	−0.30	0.0436	HK2	−0.77	0.0023
MERTK	1.00	0.0348	CD300A	0.41	0.0117	ARPC2	0.17	0.0203	AKR1B10	−0.30	0.0100	IFI27L2A	−0.77	0.0403
F13A1	0.98	0.0109	SAT1	0.41	0.0007	PCNA	0.17	0.0350	UBC	−0.31	0.0056	FGD2	−0.83	0.0048
SERPINB8	0.97	0.0282	1700017B05RIK	0.40	0.0163	UBE2J1	0.17	0.0384	AGO4	−0.32	0.0367	LY86	−0.84	0.0002
KLF10	0.95	0.0022	COTL1	0.39	0.0018	ELMO1	0.16	0.0220	APH1C	−0.35	0.0282	FCRLS	−0.85	0.0032
ATF3	0.94	0.0077	ATF4	0.39	0.0003	SEMA4D	−0.16	0.0484	EPB41L2	−0.35	0.0016	HPGD	−0.87	0.0004
HSPA1A	0.92	0.0054	SRGN	0.37	0.0237	ASAH1	−0.17	0.0333	LPCAT2	−0.35	0.0344	KLHL6	−0.95	0.0173
ARHGAP27	0.83	0.0001	ISYNA1	0.35	0.0247	B2M	−0.17	0.0416	ARHGAP11A	−0.37	0.0465	SIGLECH	−0.98	0.0005
SOCS3	0.81	0.0258	H3F3B	0.33	0.0072	LY6E	−0.19	0.0276	HEXB	−0.38	0.0003	OAS2	−0.98	0.0095
GPNMB	0.79	0.0039	PPP1R15A	0.31	0.0263	TPP1	−0.19	0.0097	CSF1R	−0.42	0.0020	P2RY12	−1.10	0.0001
PHYHD1	0.78	1.08 × 10^−5^	ARL4C	0.30	0.0029	SGPL1	−0.20	0.0388	MPEG1	−0.42	0.0088	CD74	−1.18	0.0001
CD68	0.73	0.0096	CCDC9	0.29	0.0047	IL6ST	−0.20	0.0219	GPR34	−0.43	0.0433	H2-AA	−1.55	0.0029
EGR1	0.72	0.0028	HERPUD1	0.28	0.0076	PMP22	−0.20	0.0479	CRYL1	−0.44	0.0130			
SPARC	0.71	2.21 × 10^−8^	SKI	0.28	0.0104	RRBP1	−0.20	0.0274	SALL1	−0.45	0.0173			
C3AR1	0.69	0.0154	SERPINF1	0.28	0.0375	AXL	−0.21	0.0334	RENBP	−0.46	0.0219			
SH2B2	0.68	0.0052	PTPRJ	0.27	0.0060	COMMD8	−0.21	0.0440	P2RY13	−0.48	0.0356			

**Table 2 cells-12-00745-t002:** Categorized microglia-associated genes dysregulated by ethanol exposure in the cerebellum. The microglia-associated genes identified in our data set in Table 1 with an adjusted *p* < 0.05 and Log_2_ fold change ≥ 0.25 or ≤ −0.25 were further categorized as being homeostatic or neurodegenerative, as defined by previous studies [identified in [35,36,37]].

Homeostatic	LogFC	Adj. *p*	Neurodegenerative	LogFC	Adj. *p*
MERTK	1.00	0.0348	GPX3	2.68	1.77 × 10^−9^
EGR1	0.72	0.0028	CCL2	2.44	0.0015
SLCO2B1	−0.28	0.0484	MSR1	1.34	0.0221
HEXB	−0.38	0.0003	SPP1	1.05	0.0315
CSF1R	−0.42	0.0020	GPNMB	0.79	0.0039
GPR34	−0.43	0.0433	CD68	0.73	0.0096
SALL1	−0.45	0.0173	LAIR1	0.45	0.0055
P2RY13	−0.48	0.0356	TREM2	−0.62	0.0170
KCTD12	−0.53	1.79 × 10^−5^			
Hpgds	−0.60	0.0100			
CCR5	−0.67	0.0274			
FGD2	−0.83	0.0048			
FCRLS	−0.85	0.0032			
Siglech	−0.98	0.0005			
P2RY12	−1.10	0.0001			

**Table 3 cells-12-00745-t003:** Categorized astrocyte-associated genes dysregulated by ethanol exposure in the cerebellum. Genes were identified by cross-referencing our significantly (adjusted *p* < 0.05) differentially regulated gene list with the list of 309 astrocyte-associated genes extracted from a previous study [identified in [38]] (Appendix A) using R statistical software. The astrocyte-associated genes identified in our dataset were then further categorized as being associated with acute injury, chronic neurodegenerative diseases, or pan-injury, as described in a previous study [38].

Acute Injury	LogFC	Adj. *p*	Pan Astrocytic	LogFC	Adj. *p*	Chronic Neurodegenerative Diseases	LogFC	Adj. *p*
RCAN2	0.40	0.0091	UCP2	1.20	0.0088	S1PR1	−0.33	0.0006
Lrrc58	0.31	0.0036	ATF3	0.94	0.0077	ARSK	−0.33	0.0089
ARL4C	0.30	0.0029	GPNMB	0.79	0.0039	COBL	−0.47	0.0172
PRELP	0.27	0.0368	LGALS3	0.67	0.0282			
YWHAZ	0.26	0.0014	ARHGDIB	0.54	0.0103			
DNTTIP2	0.24	0.0244	RHOJ	0.46	0.0117			
CDC42SE1	0.23	0.0082	PARP3	0.45	0.0065			
HINT1	0.22	0.0040	TIMP3	0.38	0.0216			
CARS	0.22	0.0079	AHNAK	0.33	0.0173			
IARS	0.21	0.0097	PPARGC1A	0.26	0.0276			
ARNTL	0.19	0.0240	ELOVL2	0.25	0.0113			
LRRC41	0.19	0.0461	MCL1	0.21	0.0160			
SSBP3	0.19	0.0202	AHCYL1	0.16	0.0148			
BRCC3	0.19	0.0288	B2M	−0.17	0.0416			
LRRC59	0.18	0.0391	DST	−0.21	0.0280			
UBE2F	0.18	0.0219	SQLE	−0.27	0.0246			
FARSB	0.16	0.0366	APLN	−0.28	0.0433			
CNBP	0.14	0.0482	PTPRD	−0.33	0.0006			
SGPL1	−0.20	0.0388	FLOT1	−0.33	0.0116			
AXL	−0.21	0.0334	NSDHL	−0.35	0.0137			
LAP3	−0.21	0.0321	HMGCS1	−0.43	0.0002			
SGCB	−0.21	0.0213	CTSS	−0.51	0.0005			
RNF141	−0.27	0.0039	VIM	−0.51	1.91 × 10^−5^			
SYNE1	−0.30	0.0102	IDI1	−0.55	0.0009			
POLD4	−0.34	0.0375	IFIT3	−0.75	0.0360			
PLIN2	−0.38	0.0084							
IL33	−0.91	0.0001							
IGSF1	−0.92	0.0057							

**Table 4 cells-12-00745-t004:** Categorized oligodendrocyte lineage-associated genes dysregulated by ethanol exposure in the cerebellum. Genes were identified by cross-referencing our significantly (adjusted *p* < 0.05) differentially regulated gene list with the list of OPC, COP, NFOL, MFOL and MOL-associated genes [identified in [29,30,39]] (Appendix A) using R statistical software.

OPC	LogFC		Adj. *p*	OPC		LogFC	Adj. *p*	OPC			LogFC	Adj. *p*		OPC	LogFC			Adj. *p*
PTPRN	1.03		0.0011	GNG3		0.27	0.0053	PRKCB			−0.17	0.0390		LNX1	−0.37			0.0017
SERPINA3N	0.98		0.0120	DSCAM		0.27	0.0173	DNM3			−0.18	0.0334		RSU1	−0.40			0.0007
SMOX	0.90		0.0001	NMNAT2		0.26	0.0130	DISP2			−0.18	0.0349		JAM2	−0.41			0.0006
GPNMB	0.79		0.0039	CXADR		0.25	0.0102	DDAH1			−0.20	0.0476		PHLDB1	−0.42			0.0004
SORCS1	0.60		0.0003	ABHD17B		0.25	0.0113	PCDH9			−0.22	0.0174		LBH	−0.44			0.0002
MIDN	0.42		0.0088	SCG5		0.25	0.0033	PCDH10			−0.23	0.0301		RAMP1	−0.45			0.0003
TRIL	0.39		0.0116	CHPT1		0.24	0.0110	OMG			−0.23	0.0191		EDNRB	−0.47			0.0027
HIP1	0.35		0.0003	PHACTR3		0.24	0.0278	SLC35F1			−0.24	0.0275		COBL	−0.47			0.0172
KANK1	0.33		0.0160	EHD3		0.23	0.0139	SLC22A15			−0.24	0.0188		GLTP	−0.48			0.0006
ITGAV	0.33		0.0034	DLGAP1		0.20	0.0124	PCDH17			−0.25	0.0235		GJC3	−0.48			0.0001
CALY	0.32		0.0021	ADORA1		0.20	0.0151	ADCYAP1R1			−0.25	0.0029		PTN	−0.52			0.0002
GPT2	0.31		0.0014	ZCCHC24		0.20	0.0245	SVIL			−0.26	0.0391		PLXNB3	−0.52			0.0105
CASKIN2	0.31		0.0163	PTPRE		0.20	0.0168	KLHL5			−0.27	0.0075		MMP15	−0.56			0.0239
KCNK3	0.30		0.0130	RAB31		0.19	0.0231	GRIA4			−0.29	0.0018		RCN1	−0.65			0.0103
NCALD	0.30		0.0041	NELL2		0.19	0.0125	SERINC5			−0.30	0.0016		RLBP1	−0.78			0.0021
LRRFIP1	0.29		0.0024	GNPTG		0.18	0.0202	KLHL13			−0.31	0.0113		EMID1	−0.84			0.0013
CAV2	0.28		0.0473	GAD1		0.15	0.0246	CSPG5			−0.34	0.0086		PLLP	−1.11			0.0001
SDC3	0.28		0.0411	NOVA1		−0.16	0.0402	GNB4			−0.35	0.0008						
**COP**			**LogFC**			**Adj. *p***				**NFOL**				**LogFC**		**Adj. *p***		
TIMP4			0.42			0.0001				H2-AB1				−1.38		0.0007		
SEZ6L			0.40			0.0005				SEMA4D				−0.16		0.0484		
SIRT2			−0.16			0.0479												
SLC44A1			−0.18			0.0460												
EDIL3			−0.20			0.0247												
S100B			−0.24			0.0080												
BCAS1			−0.28			0.0412												
CNP			−0.30			0.0066												
GPR17			−0.33			0.0116												
EPB41L2			−0.35			0.0016												
LIMS2			−0.38			0.0468												
ENPP6			−0.53			0.0036												
**MFOL**	**LogFC**	**Adj. *p***	**MFOL**	**LogFC**	**Adj. *p***	**MFOL**		**LogFC**	**Adj. *p***	**MFOL**			**LogFC**	**Adj. *p***	**MOL**		**LogFC**	**Adj. *p***
APOD	1.66	0.0001	LAP3	−0.21	0.0321	SEPTIN4		−0.35	0.0005	UGT8A			−1.16	0.0020	NINJ2		−1.88	0.0005
HSPA1A	0.92	0.0054	ATP8A1	−0.21	0.0091	ERMN		−0.37	0.0346	SERPINB1A			−1.28	3.22 × 10^−5^	KLK6		−1.04	0.0016
ADIPOR2	0.90	0.0018	SCCPDH	−0.21	0.0377	MAG		−0.39	0.0346	OPALIN			−2.33	6.07 × 10^−7^				
GLUL	0.79	0.0010	FGFR2	−0.21	0.0362	QDPR		−0.41	0.0029									
PIM3	0.64	0.0005	FNBP1	−0.21	0.0116	PHLDB1		−0.42	0.0004									
KLF13	0.53	0.0001	CCP110	−0.22	0.0142	MAP6D1		−0.43	0.0002									
HAPLN2	0.42	0.0267	DIP2A	−0.22	0.0113	CRYAB		−0.43	0.0445									
TUBB4A	0.41	0.0036	PCDH9	−0.22	0.0174	ABCA8A		−0.46	0.0122									
FTH1	0.39	0.0054	TPST1	−0.23	0.0279	GNG11		−0.46	0.0049									
KNDC1	0.39	0.0335	DOCK10	−0.23	0.0350	NIPA1		−0.47	0.0001									
SLC38A2	0.34	0.0003	CNTN2	−0.23	0.0218	GLTP		−0.48	0.0006									
SLC20A2	0.30	0.0013	TULP4	−0.23	0.0022	GPR37		−0.48	0.0005									
CFL2	0.28	0.0040	OMG	−0.23	0.0191	GJC3		−0.48	0.0001									
ZDHHC20	0.24	0.0249	EPS15	−0.24	0.0189	CAR2		−0.50	0.0010									
NUDT4	0.24	0.0047	ARAP2	−0.24	0.0130	PRR5L		−0.50	0.0043									
LPGAT1	0.21	0.0097	AATK	−0.25	0.0321	ANO4		−0.50	0.0010									
PAK1	0.21	0.0071	SEMA6D	−0.25	0.0062	ARSG		−0.52	0.0029									
TMOD2	0.20	0.0160	KCNA6	−0.27	0.0047	PLXNB3		−0.52	0.0105									
GPX4	0.20	0.0175	GATM	−0.27	0.0091	1700047M11RIK		−0.53	0.0012									
PSAT1	0.19	0.0409	BCAS1	−0.28	0.0412	LPAR1		−0.54	0.0012									
PCNP	0.18	0.0231	S1PR5	−0.29	0.0214	TMEM88B		−0.56	0.0002									
CDC37L1	0.16	0.0424	GRM3	−0.29	0.0346	CMTM5		−0.59	0.0017									
ATP6AP2	0.16	0.0309	EPHB1	−0.29	0.0059	FA2H		−0.67	0.0004									
DENND5A	−0.16	0.0239	UNC5B	−0.29	0.0226	ASPA		−0.67	0.0001									
ACOT7	−0.17	0.0496	TMEFF1	−0.30	0.0304	HHIP		−0.73	0.0033									
MYO6	−0.17	0.0271	SERINC5	−0.30	0.0016	TMEM125		−0.75	0.0102									
SLC44A1	−0.18	0.0460	CNP	−0.30	0.0066	SOX2OT		−0.85	0.0052									
SORT1	−0.18	0.0127	TTYH2	−0.31	0.0053	PPP1R14A		−0.86	0.0011									
DNM3	−0.18	0.0334	TPPP	−0.32	0.0026	MOG		−0.86	0.0010									
ANK3	−0.19	0.0130	TRIM59	−0.33	0.0334	PDLIM2		−0.87	0.0014									
YPEL2	−0.20	0.0410	REEP3	−0.33	0.0022	IL33		−0.91	0.0001									
EDIL3	−0.20	0.0247	PTPRD	−0.33	0.0006	PRR18		−0.91	0.0003									
KCNJ10	−0.20	0.0348	PACS2	−0.34	0.0008	PLP1		−1.07	5.01 × 10^−7^									
WNK1	−0.20	0.0039	DPY19L1	−0.34	0.0012	PLLP		−1.11	0.0001									
DST	−0.21	0.0280	TSPAN2	−0.35	0.0008	GJC2		−1.11	0.0043									

**Table 5 cells-12-00745-t005:** Tabular descriptions of the top canonical pathway categories, including pathway names, p-values, and indicated molecules. Qiagen Ingenuity Pathway Analysis (IPA) software was utilized to assess the top canonical pathways altered by ethanol exposure using the “cerebellum” selected analysis settings. All analyses were restricted to genes with an adjusted *p* < 0.05 and Log_2_ fold change ≥0.5 or ≤−0.5.

Pathway Category	Pathway Name	*p*-Value	Molecules
Generation of precursor metabolites and energy	Glycerol-3-phosphate shuttle	0.0469	GPD1
Pathogen-influenced signaling	LPS/IL-1 mediated inhibition of RXR function	0.0400	CHST7, GSTM5, IL33, RARA, SMOX, SREBF1
Cellular immune response	Granulocyte adhesion and diapedesis	0.0303	C5AR1, IL33, SDC4, SELPLG
Degradation/utilization/assimilation	Tryptophan degradation X	0.0481	AKR1B10, SMOX
	Glycerol degradation I	0.0469	GPD1
	Dopamine degradation	0.0368	SMOX, Sult1a1
	Acetone degradation I (to Methylglyoxal)	0.0268	AKR1B10, CYP51A1
	Spermine and spermidine degradation I	0.0237	SMOX
Cellular stress and injury	Intrinsic prothrombin activation pathway	0.0481	COL5A3, KLK6
	GP6 signaling pathway	0.0388	COL16A1, COL27A1, COL5A1, COL5A3
	Wound-healing signaling pathway	0.0288	COL16A1, COL27A1, COL5A1, COL5A3, IL33, VIM
	Coagulation system	0.0181	F3, VWF
	Osteoarthritis pathway	0.0163	ANXA2, FGFR3, GREM1, HES1, HTRA1, SDC4, SPP1
	Apelin liver signaling pathway	0.0059	AGT, COL5A3, EDN1
	Pulomary fibrosis idiopathic signaling pathway	0.0015	CCN2, COL16A1, COL27A1, COL5A1, COL5A3, EDN1, EGR1, FGFR3, HES1, LPAR1, VIM
Biosynthesis	Trans, trans-faresyl diphosphate biosynthesis	0.0469	IDI1
	Cholesterol biosynthesis III (via desmosterol)	0.0316	CYP51A1, MSMO1
	Glutamine biosynthesis I	0.0237	GLUL
	Superpathway of citrulline metabolism	0.0223	ASL, PRODH
	Γ-linolenate biosynthesis II	0.0181	FADS1, FADS2
	Superpathway of geranylgeranyldiphosphate biosynthesis I (via mevalonate)	0.0143	ACAT2, IDI1
	Mevalonate pathway I	0.0109	ACAT2, IDI1
	Zymosterol biosynthesis	0.0054	CYP51A1, MSMO1
	Superpathway of cholesterol biosynthesis	0.0011	ACAT2, CYP51A1, IDI1, MSMO1
Disease-specific pathway	Osteoarthritis pathway	0.0163	ANXA2, FGFR3, GREM1, HES1, HTRA1, SDC4, SPP1
	Pathogen-induced cytokine storm signaling pathway	0.0111	COL16A1, COL27A1, COL5A1, COL5A3, DHX58, IL33, SOCS3
	Hepatic fibrosis/hepatic stellate cell activation	0.0040	AGT, CCN2, COL16A1, COL27A1, COL5A1, COL5A3, EDN1
	Pulomary fibrosis idiopathic signaling pathway	0.0015	CCN2, COL16A1, COL27A1, COL5A1, COL5A3, EDN1, EGR1, FGFR3, HES1, LPAR1, VIM
	Atherosclerosis signaling	0.0005	APOD, COL5A3, F3, IL33, SELPLG, TNFRSF12A
Cardiovascular signaling	Intrinsic prothrombin activation pathway	0.0481	COL5A3, KLK6
	Atherosclerosis signaling	0.0005	APOD, COL5A3, F3, IL33, SELPLG, TNFRSF12A
Nuclear receptor signaling	LPS/IL-1 mediated inhibition of RXR function	0.0400	CHST7, GSTM5, IL33, RARA, SMOX, SREBF1
	LXR/RXR activation	0.0103	AGT, APOD, CYP51A1, IL33, SREBF1
	FXR/RXR activation	0.0064	AGT, APOD, IL33, RARA, SREBF1
	VDR/RXR activation	0.0002	CDKN1A, HES1, IGFBP1, KLF4, KLK6, SPP1
Ingenuity toxicity list pathways	LPS/IL-1 mediated inhibition of RXR function	0.0400	CHST7, GSTM5, IL33, RARA, SMOX, SREBF1
	LXR/RXR activation	0.0103	AGT, APOD, CYP51A1, IL33, SREBF1
	FXR/RXR activation	0.0064	AGT, APOD, IL33, RARA, SREBF1
	Hepatic fibrosis/hepatic stellate cell activation	0.0040	AGT, CCN2, COL16A1, COL27A1, COL5A1, COL5A3, EDN1
	VDR/RXR activation	0.0002	CDKN1A, HES1, IGFBP1, KLF4, KLK6, SPP1

**Table 6 cells-12-00745-t006:** Tabular descriptions of the disease and biological function categories, including annotation, *p*-value, and indicated molecules. Qiagen Ingenuity Pathway Analysis (IPA) software was utilized to assess the top diseases and biological functions altered by ethanol exposure using the “cerebellum” selected analysis settings. All analyses were restricted to genes with an adjusted *p* < 0.05 and Log_2_ fold change ≥ 0.5 or ≤−0.5.

Categories	Disease or Function Annotation	*p*-Value	Molecules
Nervous system development and function	Myelination	2.88 × 10^−6^	ASPA, FGFR3, GJB6, GJC2, HPGDS
Nervous system development and function, tissue Morphology	Quantity of oligodendrocytes	0.000125	FGFR3, GJB6, GJC2
Cell-to-cell signaling and interaction	Coupling of oligodendrocytes	0.000556	GJB6, GJC2
Cell morphology, cellular assembly and organization, nervous system development and function, tissue morphology	Thickness of myelin sheath	0.000556	GJB6, GJC2
Cell-to-cell signaling and interaction	Coupling of astrocytes	0.000556	GJB6, GJC2
Cellular assembly and organization	Formation of vacuole	0.00164	GJB6, GJC2
Developmental disorder, nervous system development and function, neurological disease, organismal injury and abnormalities	Demyelination of cerebellum	0.0053	ASPA, HPGDS
Cell death and survival, cellular compromise, neurological disease, organismal injury and abnormalities, tissue morphology	Neurodegeneration of axons	0.0053	ASPA, SPTSSB
Tissue morphology	Quantity of cells	0.00738	ARSG, ASPA, FGFR3, GJB6, GJC2, NRN1
Cell-to-cell signaling and interaction, hematological system development and function, immune cell trafficking, inflammatory response, nervous system development and function	Activation of microglia	0.00783	GJB6, GJC2
Nervous system development and function	Morphology of nervous system	0.011	ARSG, FA2H, GJB6, GJC2, MERTK, PLP1, RARA, TBATA, UGT8, ZIC4
Nervous system development and function, tissue morphology	Morphology of nervous tissue	0.0126	ARSG, FA2H, GJB6, GJC2, PLP1, TBATA, UGT8
Cellular compromise, neurological disease, organismal injury and abnormalities	Damage of axons	0.0236	SOCS3
Cell-to-cell signaling and interaction, nervous system development and function	Synaptic transmission of Bergmann glia	0.0236	SLC1A6
Embryonic development, nervous system development and function, organ development, organismal development, tissue development	Delay in myelination of cerebellum	0.0236	FGFR3
Cardiovascular system development and function, nervous system development and function, organ morphology, tissue morphology	Permeability of blood–brain barrier	0.0236	MOG
Nervous system development and function, neurological disease, organismal injury and abnormalities	Abnormal morphology of nervous system	0.0314	ARSG, FA2H, MERTK, PLP1, RARA, TBATA, UGT8, ZIC4
Cellular assembly and organization, cellular function and maintenance, nervous system development and function, tissue morphology	Quantity of dendrites	0.0467	NRN1
Neurological disease, organismal injury and abnormalities, psychological disorders	Spongy degeneration of central nervous system of white matter	0.0467	ASPA
Neurological disease, organismal injury and abnormalities	Astrocytosis of cerebellum	0.0467	HPGDS

## Data Availability

The data discussed in this publication have been deposited in NCBI’s Gene Expression Omnibus [88] and are accessible through GEO Series accession number GSE222445 (https://www.ncbi.nlm.nih.gov/geo/query/acc.cgi?acc=GSE222445, accessed on 24 February 2023).

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
