# Peer review of "Cerebellar Transcriptomic Analysis in a Chronic plus Binge Mouse Model of Alcohol Use Disorder Demonstrates Ethanol-Induced Neuroinflammation and Altered Glial Gene Expression"

_cells, 2023, doi:10.3390/cells12050745_

Round 1

Reviewer 1 Report

The paper describes outcomes of cerebellum RNAseq using a model of chronic + binge ethanol exposure in adult male mice. Analysis focused on effects on glia and support the idea that alcohol changes glial phenotypes consistent with inflammatory responses, although any accompanying pathological changes have not been shown in mouse brain using this model.

Add sample size and blood ethanol concentration information to Methods

Were BECs at or near 0 at time of tissue collection? If not, or if this is not known it should be included as a limitation.

If this is a separate analysis of a subset of animals described reported in the 2022 paper from the Drew lab that should be made clear.

In addition to p values for pathways, it would be useful to have the number of genes in each pathway in addition to the ID of those that are significantly different.

Limitations should be included in the discussion, including that the study is only in males, that specific gene effects in this same model were also described in 2022 and that changes seen here do not replicate those, that some of the pathways only have 1 or 2 genes and those are also represented in other pathways, and that this is a study of very acute effects.

Table 2 – check adj. p value for AHNAK

Author Response

Response to Reviewer 1 Comments

General Comments:  The paper describes outcomes of cerebellum RNAseq using a model of chronic + binge ethanol exposure in adult male mice. Analysis focused on effects on glia and support the idea that alcohol changes glial phenotypes consistent with inflammatory responses, although any accompanying pathological changes have not been shown in mouse brain using this model.

Point 1:  Add sample size and blood ethanol concentration information to Methods

Response 1:  Sample size and BEC information has been added to the Animals subsection of the Materials and Methods.

Point 2:  Were BECs at or near 0 at time of tissue collection? If not, or if this is not known it should be included as a limitation.

Response 2:  We have not formally evaluated BECs at the time of tissue harvest in this chronic plus binge model. We suspect that BECs were at or near 0 at the time of tissue harvest since we have shown that BECs are near 0 24h following a single bolus (binge) of ethanol. However, these animals were not treated chronically with ethanol prior to the bolus. Thus, we can not say definitively that BECs were near 0 at the time of harvest in our chronic plus binge model. We have added this information to the Animals subsection of the Materials and Methods as a limitation as suggested by the reviewer.

Point 3:  If this is a separate analysis of a subset of animals described reported in the 2022 paper from the Drew lab that should be made clear.

Response 3:  The animals included in the current study are unique to the current manuscript and were not included in any previously published studies.  We have clarified this distinction in the Discussion.

Point 4:  In addition to p values for pathways, it would be useful to have the number of genes in each pathway in addition to the ID of those that are significantly different.

Response 4:  We have added the number of genes in each pathway in addition to the ID of those that were significantly different as suggested by the reviewer. In order to accomplish this change, we created a new table (now Table #4 – Tabular description of the top cannonical pathway catagories), that is similar to the previous Table #4 (now Table #5 – Tabular descriptions of the diseases and biological functions categories).  The new Table 4 has been inserted into the marked pre-layout copy of the main document.  An updated word document containing all Tables has also been provided via the online portal.

Point 5:  Limitations should be included in the discussion, including that the study is only in males, that specific gene effects in this same model were also described in 2022 and that changes seen here do not replicate those, that some of the pathways only have 1 or 2 genes and those are also represented in other pathways, and that this is a study of very acute effects.

Response 5:  We have added the limitations noted by the reviewer to the Discussion.

Point 6:  Table 2 – check adj. p value for AHNAK

Response 6:  The adj. p value for AHNAK has been corrected in Table 2 of the manuscript and a revised copy of Table 2 has been provided via the online portal.

Reviewer 2 Report

This manuscript describes gene expression changes in the cerebellum after mice were treated with the “chronic plus binge” model of alcohol exposure. RNA-Seq was performed on bulk cerebellum tissue and then gene expression changes categorized into cell types based on publicly available published single cell datasets. The authors found overall decreases in microglial genes associated with homeostasis and increases in microglial genes associated with neurodegeneration, as well as increased expression of astrocyte expressed genes associated with acute injury. Although this is a descriptive study without validation of the findings by qPCR, the categorization of genes by cell type and RNA-Seq of the cerebellum is unique and should be published. Minor corrections need to be made to the manuscript.

1.     Please note that since the introduction of DSM-5, the proper term is alcohol use disorder and not alcohol use disorders. Please correct throughput manuscript, including in title.

2.     Was a specific lobe or region of the cerebellum dissected for the RNA-Seq study? If so, please indicate this in the Materials and Methods.

3.     There are locations in the manuscript pdf where text appears to be missing: p. 5, after line 162, there is no paragraph describing the results in Table 2; p. 13, after line 312, sentence is not complete.

4.     It’s hard to read the axis labels and legend text on the graphs for Fig. 5C-E. Please increase font size.

5.     Line 36 “American’s” should be “Americans”

6.      P. 12, line 290, add the word “are” after astrocytes.

7.     The discussion should include a limitation of the study in that only male mice were used.

Author Response

Response to Reviewer 2 Comments

General Comments:  This manuscript describes gene expression changes in the cerebellum after mice were treated with the “chronic plus binge” model of alcohol exposure. RNA-Seq was performed on bulk cerebellum tissue and then gene expression changes categorized into cell types based on publicly available published single cell datasets. The authors found overall decreases in microglial genes associated with homeostasis and increases in microglial genes associated with neurodegeneration, as well as increased expression of astrocyte expressed genes associated with acute injury. Although this is a descriptive study without validation of the findings by qPCR, the categorization of genes by cell type and RNA-Seq of the cerebellum is unique and should be published. Minor corrections need to be made to the manuscript.

Point 1:  Please note that since the introduction of DSM-5, the proper term is alcohol use disorder and not alcohol use disorders. Please correct throughput manuscript, including in title.

Response 1:  We have corrected as suggested in the Title, Abstract (line 13), and line 35.  No other instances were found.

Point 2:  Was a specific lobe or region of the cerebellum dissected for the RNA-Seq study? If so, please indicate this in the Materials and Methods.

Response 2:  We have added clarification to the Materials and Methods (lines 89-97) indicating that one whole cerebellar hemisphere was used for RNA isolation from each experimental animal.

Point 3:  There are locations in the manuscript pdf where text appears to be missing: p. 5, after line 162, there is no paragraph describing the results in Table 2; p. 13, after line 312, sentence is not complete.

Response 3:  Thank you for pointing this out.  A whole paragraph appears to be missing following line 162, which may be the result of a layout error.  We have added the following paragraph which was included with the original submission:

                “Similar to microglia, we utilized scRNA-seq data to compose a list of 309 astrocyte associated genes (Supplemental Table S2) [37]. From this list we identified 56 astrocyte associated genes that were differentially expressed in response to ethanol in our current study. We then characterized these transcripts as being associated with an astrocyte phenotype common to acute injury, chronic neurodegenerative diseases, or pan-injury (Table 2), the last of which includes genes associated with both acute injury and chronic neurodegenerative diseases phenotypes  [37].”

We have corrected the sentence beginning on line 312.

Point 4:  It’s hard to read the axis labels and legend text on the graphs for Fig. 5C-E. Please increase font size.

Response 4:  We have increased the axis labels and text for Fig. 5C-E in addition to Fig. 3B-C and Fig. 4B-C.  We removed the legends since this information was duplicitous. The updated figures have been inserted into the marked copy of the pre-layout and have been updated via the online portal.

Point 5:  Line 36 “American’s” should be “Americans”

Response 5:  “American’s” has been corrected to “Americans”.

Point 6:  P. 12, line 290, add the word “are” after astrocytes.

Response 6:  The word “are” has been added.

Point 7:  The discussion should include a limitation of the study in that only male mice were used.

Response 7:  A limitation statement concerning only use of males has been added to the discussion section.